# Genome-Wide Identification and Characterization of the *NF-YA* Gene Family and Its Expression in Response to Different Nitrogen Forms in *Populus × canescens*

**DOI:** 10.3390/ijms231911217

**Published:** 2022-09-23

**Authors:** Jing Zhou, Lingyu Yang, Xin Chen, Mengyan Zhou, Wenguang Shi, Shurong Deng, Zhibin Luo

**Affiliations:** State Key Laboratory of Tree Genetics and Breeding, Key Laboratory of Silviculture of the National Forestry and Grassland Administration, Research Institute of Forestry, Chinese Academy of Forestry, Beijing 100091, China

**Keywords:** *Populus* × *canescens*, NF-YA, gene family, expression analysis, nitrogen

## Abstract

The *NF-YA* gene family is a class of conserved transcription factors that play important roles in plant growth and development and the response to abiotic stress. Poplar is a model organism for studying the rapid growth of woody plants that need to consume many nutrients. However, studies on the response of the *NF-YA* gene family to nitrogen in woody plants are limited. In this study, we conducted a systematic and comprehensive bioinformatic analysis of the *NF-YA* gene family based on *Populus* × *canescens* genomic data. A total of 13 *PcNF-YA* genes were identified and mapped to 6 chromosomes. According to the amino acid sequence characteristics and genetic structure of the NF-YA domains, the PcNF-YAs were divided into five clades. Gene duplication analysis revealed five pairs of replicated fragments and one pair of tandem duplicates in 13 *PcNF-YA* genes. The *PcNF-YA* gene promoter region is rich in different *cis*-acting regulatory elements, among which MYB and MYC elements are the most abundant. Among the 13 *PcNF-YA* genes, 9 contained binding sites for *P.* × *canescens* miR169s. In addition, RT-qPCR data from the roots, wood, leaves and bark of *P.* × *canescens* showed different spatial expression profiles of *PcNF-YA* genes. Transcriptome data and RT-qPCR analysis showed that the expression of *PcNF-YA* genes was altered by treatment with different nitrogen forms. Furthermore, the functions of *PcNF-YA* genes in transgenic poplar were analyzed, and the potential roles of *PcNF-YA* genes in the response of poplar roots to different nitrogen forms were revealed, indicating that these genes regulate root growth and development.

## 1. Introduction

Nuclear transcription factor Y (NF-Y) proteins can bind to CCAAT-boxes in promoter sequences [1]. They generally consist of three subunits: NF-YA, NF-YB, and NF-YC. In plants, each NF-Y subunit has evolved as a polygenic coding family with a large number of heterotrimeric combinations [2]. The conserved core functional regions of plant NF-YA proteins consist of 53 amino acids and contain two conserved domains: the NF-YB/C interaction and DNA binding domains. The NF-YB/C interaction domain is responsible for binding to NF-YB and NF-YC, and the DNA binding domains bind CCAAT elements in gene promoters in a highly specific manner to regulate transcription [3]. Despite the abundance of NF-YB and NF-YC subunits, NF-YA restricts heterotrimer formation and subsequent DNA binding [4]. Therefore, the roles of NF-YA transcription factors (TFs) are particularly critical.

The *NF-YA* gene family contains one or two members in yeast and mammals but has expanded significantly in plants [4]. To date, researchers have reported on the NF-YA family in several plants, including *Arabidopsis thaliana* [5,6,7,8,9], wheat [10], *Prunus persica* L [11], *Pinus tabuliformis* [12], *Citrus sinensis* [13], *Ricinus communis* [14], and *Solanum tuberosum* L. [15], in which they identified 10, 10, 6, 9, 6, 6 and 10 *NF-YA* family members, respectively. Additionally, *NF-YA* genes are target genes of miR169s, which are widely involved in seed development, root architecture, nitrogen fixation, flowering, biotic and abiotic stress responses and other processes and are considered key regulators of plant growth and development [5,16,17,18,19,20,21]. For example, *AtNF-YA2* regulates apical meristem development and lateral root growth and development in *A. thaliana* [5]. *AtNF-YA2* is also involved in regulating the growth arrest of *A. thaliana* after germination under salt stress [6]. *AtNF-YA2* and *AtNF-YA10* regulate *A. thaliana* leaf growth and development through the auxin signaling pathway [7]. *AtNF-YA3* and *AtNF-YA8* play roles in *A. thaliana* embryogenesis [8]. *AtNF-YA5* is involved in the response to drought stress [9].

Poplar is one of the main fast-growing, high-yield wood species cultivated in China [22]. Due to its rapid growth, poplar exhibits high nutrient consumption levels, especially for nitrogen, one of the essential mineral element for plant growth [22]. Therefore, the nitrogen absorption and utilization efficiency of tree roots directly affects the growth, development and yield of fast-growing and high-yield timber forest species [23,24]. Previous studies have shown that different root zones of *Populus* × *canescens* exhibit different uptake efficiencies for different nitrogen forms [25,26]. In addition, the *P.* × *canescens* genome has been sequenced and functionally annotated [27]. This study provides basic sequence information for the study of *NF-YA* family genes in *P.* × *canescens*. In addition, *NF-YA* genes have been reported to be involved in the nitrogen response in herbaceous plants [21,28,29]. Overexpression of *NF-YAs* can regulate the expression of *nitrate transporters* (*NRTs*) in wheat and *Oryza sativa* [21,28]. The nitrate uptake of transgenic wheat roots is increased, and lateral root growth is promoted [21]. Previous studies have shown that *NF-YA* genes are involved in the response of herbaceous plants to nitrogen, especially at low nitrogen levels. However, it is worth exploring whether *NF-YA* genes respond to different nitrogen forms that can be directly absorbed and utilized by woody plants.

In this study, the *NF-YA* gene family members of *P.* × *canescens* were identified at the whole-genome level. Bioinformatics analysis software was used to analyze the physical and chemical properties, gene structures and domains, evolutionary relationships, chromosome localization and replication modes, miRNA targets and *cis*-acting promoter elements of the gene family. This study provides basic information for further studying the evolutionary relationships of woody plant *NF-YA* gene family members. Furthermore, the tissue expression patterns of the *PcNF-YA* family genes in *P.* × *canescens* and their expression patterns under treatment with different nitrogen forms were studied. The functions of *PcNF-YA* genes were further identified in transgenic poplar, which confirmed that *PcNF-YA* genes regulated the growth and development of woody plants in response to nitrate.

## 2. Results

### 2.1. Identification of NF-YA Genes and Characterization of NF-YA Proteins in P. × canescens

To identify the *NF-YA* gene family members of *P. × canescens*, a hidden Markov model (Pfam number: PF02045) and the BLASTP method were used to conduct a genome-wide search of the reference genome of *P. × canescens*. After confirming the presence of specific conserved CBF_NF-YA domains and NF-YB/C interaction domains, thirteen putative *PcNF-YA* genes were finally obtained and were found to be distributed on 6 chromosomes of *P. × canescens* (Chr1, 6, 9, 11, 7, and 18) (Table 1). The PcNF-YA proteins were analyzed with the online software Expasy. The proteins encoded by the *PcNF-YA* genes were 53–377 amino acids in length and had 1–6 exons. Their molecular weights ranged from 5.80 to 41.32 kDa, and their pI values ranged from 6.35 to 11.47 (Table 1). Since NF-YAs are TFs, subcellular localization analysis showed that the PcNF-YAs were located in the nucleus (Table 1).

### 2.2. Phylogenetic and Multiple Sepuence Alignment Analyses and Tertiary Structure Prediction of PcNF-YAs

To explore the evolutionary relationships of NF-YA proteins in other plants, a phylogenetic tree of the NF-YA proteins from *P. × canescens* (Pc), *P. trichocarpa* (Ptr), *A. thaliana* (At), *O. sativa* (Os), *Pinus tabuliformis* (*Pita*) and *Prunus persica* L (Pp) was constructed by using MEGAX software with neighborhood connection (neighbor-joining, NJ) standards (Appendix A). The results showed that the PcNF-YA proteins could be divided into five categories, designated Clades I–V (Figure 1a). The clade containing the most PcNF-YA protein members was Clade II, which contained four PcNF-YA proteins (PcNF-YA 1/3/5/11). The second largest was Clade III, which contained three PcNF-YA proteins (PcNF-YA 4/9/12). Each of the other three clades contained two PcNF-YA proteins (Figure 1a). To obtain further information about PcNF-YA proteins in *P. × canescens*, we extracted conserved NF-YA protein sequences (approximately 53 amino acids) from *P. × canescens, P. trichocarpa*, *A. thaliana*, *O. sativa*, *Pinus tabuliformis* and *Prunus persica* and used them for multiple sequence alignment. As observed in other species, the conserved region of the PcNF-YAs contained a domain that interacts with the NF-YB/C subunit and a highly conserved DNA binding sequence that recognizes the CCAAT binding motif (Appendix A). Furthermore, the tertiary structure of the conserved NF-YA protein sequences of *P. × canescens* and the five other species was mapped using Phyre^2^ software. The results showed that the conserved NF-YA protein sequences of *P. × canescens* and the other species were composed of two helices and one linker (Appendix A).

### 2.3. Gene Structure and Protein Motif Analyses of PcNF-YAs

Since gene structure diversity can reflect the evolution of gene families, the exon-intron structures of the *P. × canescens PcNF-YA* genes were analyzed using the Gene Structure Display Server (GSDS). The results showed that a 5’ UTR and 3’ UTR existed in all *PcNF-YA* gene sequences. The number of exons ranged from 1 to 6 (Figure 1b,c and Table 1), with *PcNF-YA13* containing only one exon and no introns, while *PcNF-YA1* and *PcNF-YA3* contained 6 exons and 5 introns (maximum numbers), respectively. The intron-exon structures of the genes in the same clade were similar, but those of genes in different clades were different. To further understand the conservation and diversity of the PcNF-YA proteins, the MEME program was used to evaluate the conserved motifs of the PcNF-YA proteins. A total of 10 motifs (Appendix A) were identified. Among them, most PcNF-YA proteins contained motif 1, motif 2, motif 3, motif 4 and motif 5; motif 6 and motif 9 existed only in Clade I; motif 7 and motif 10 existed only in Clade III; and motif 8 existed only in Clade IV (Figure 1d).

### 2.4. Chromosome Localization, Gene Duplication and Synteny Analysis of PcNF-YA Genes

According to the obtained gene localization information, the 13 *PcNF-YA* genes were unevenly distributed on 6 chromosomes of *P. × canescens*. The detailed chromosomal locations are shown in Figure 2a. The number of *PcNF-YA* genes on chromosomes 1 and 6 was the highest, with three on each chromosome, while chromosomes 9, 11 and 16 each contained two *PcNF-YA* genes, and chromosome 18 contained one *PcNF-YA* gene.

To further study the replication events of the *PcNF-YA* gene family in *P. × canescens*, MCScanX was used to analyze *PcNF-YA* gene fragment replication and tandem duplication events (Figure 2b). The results showed that there were five pairs of fragment replicates on the chromosomes of *P. × canescens*: *PcNF-YA2* (Chr 18) and *PcNF-YA6* (Chr 6), *PcNF-YA5* (Chr 9) and *PcNF-YA11* (Chr 1), *PcNF-YA7* (Chr 6) and *PcNF-YA8* (Chr 16), *PcNF-YA4* (Chr 1) and *PcNF-YA12* (Chr 11), and *PcNF-YA9* (Chr 1) and *PcNF-YA12* (Chr 11). In addition, *PcNF-YA9* and *PcNF-YA12* were aggregated into tandem duplication events on Chr 11 (Figure 2b).

To further explore the interspecific evolutionary behavior of the *PcNF-YA* gene family in *P. × canescens*, we analyzed the collinearity maps of *P. × canescens*, *A. thaliana*, *O. sativa* and *Prunus persica* and revealed the interspecific expansion process of *PcNF-YA* family members (Figure 2c and Appendix A). The results showed that the *PcNF-YA* genes were collinear with genes in *A. thaliana*, *O. sativa* and *Prunus persica*. The number of collinear gene pairs varied among different species. Eight *NF-YA* genes showed collinearity in the genomes of *P. × canescens* and *A. thaliana*, while there were 6 collinear gene pairs between *P. × canescens* and *O. sativa*, and there were 12 pairs (the largest number) of collinear genes in *P. × canescens* and *Prunus persica* (Figure 2c and Appendix A).

### 2.5. Analysis of Cis-Acting Regulatory Elements of PcNF-YA Genes

To explore the expression and transcriptional regulation characteristics of *PcNF-YA* gene family members, we obtained the sequence of the 2 kb region upstream of each gene-coding region from the *P. × canescens* genomic database and analyzed the promoter sequences of the 13 *PcNF-YA* genes using the PlantCARE database. The results showed that in addition to the two core components (CAAT-box and TATA-box), the 13 promoter sequences of the *PcNF-YAs* contained 30 *cis*-regulatory elements, which responded to phytohormones and abiotic and biotic stress and participated in plant growth and development (Figure 3). Among these elements, MYB (52) and MYC (39), which are important functional elements under abiotic and biotic stress, accounted for the highest proportion. Additionally, the CGTCA and TGACG motifs (methyl jasmonate (MeJA)-induced elements) and an abscisic acid (ABA)-response element (ABRE) were widely present in most of the *PcNF-YA* genes (Figure 3), among which *PcNF-YA10* contained 10 MeJA-induced elements, and *PcNF-YA2* contained 7 ABERs, accounting for the largest proportions.

### 2.6. miR169 Family Members in P. × canescens and Their Target Sites in PcNF-YA Genes

To explore the miR169-mediated posttranscriptional regulation of *PcNF-YA* genes, we identified miR169 family members in the roots of *P. × canescens* (Appendix A), which was based on previous studies on the sequencing of small RNAs [30]. BLAST searches of the mature miR169 sequences and *PcNF-YA* cDNA sequences revealed that nine *PcNF-YA* genes (*PcNF-YA1/2/3/5/6/7/8/10*/*11*) possessed miR169 binding sites (Figure 4), among which the miR169 binding sites of the *PcNF-YA1/3* genes were located in the coding sequence (CDS), and the miR169 binding sites of the other seven *PcNF-YA* genes were all located in the 3’ UTR (Figure 4).

### 2.7. Expression Patterns of PcNF-YA Genes in Different Tissues

To understand the physiological functions of the *PcNF-YA* genes in *P.* × *canescens*, the expression patterns of the *PcNF-YA* genes in different tissues, such as the roots, wood, leaves and bark, were identified by real-time quantitative PCR (RT-qPCR) (Figure 5a and Appendix A). The results showed that the expression patterns of the 13 *PcNF-YA* genes varied across tissues of *P.* × *canescens*. Six of the *PcNF-YA* genes were highly expressed in all tissues. The relative expression levels of *PcNF-YA2* and *PcNF-YA6* in Clade IV were highest in the roots, and the relative expression levels of *PcNF-YA1* and *PcNF-YA3* in Clade II were highest in the leaves. The relative expression levels of *PcNF-YA*9 and *PcNF-YA*12 in Clade III were highest in the wood. The expression levels of the other seven *PcNF-YA* family members were relatively low in all the tissues. Overall, most of the *PcNF-YA* family members in the same clade showed similar expression patterns in different tissues of *P.* × *canescens* (Figure 5a and Appendix A).

### 2.8. Expression Pattern Analysis of PcNF-YA Genes in Response to Different Nitrogen Forms

Based on the transcriptome data of *P.* × *canescens* obtained in our previous study [30], the expression patterns of the *PcNF-YA* genes in the presence of different nitrogen forms were analyzed (Figure 5b and Appendix A). The results showed that *PcNF-YA4* and *PcNF-YA12* were highly expressed after 21 days of treatment with the three tested nitrogen forms. The expression of *PcNF-YA9* was significantly upregulated only after ammonium nitrate treatment and significantly downregulated after nitrate treatment. The expression of *PcNF-YA7* was significantly upregulated after ammonium treatment and significantly downregulated after ammonium nitrate treatment. Moreover, the expression levels of *PcNF-YA2*, *PcNF-YA5* and *PcNF-YA11* were significantly downregulated after nitrate treatment. However, the expression levels of *PcNF-YA1*, *PcNF-YA3* and *PcNF-YA6* were significantly downregulated after ammonium treatment (Figure 5b).

Furthermore, RT-qPCR was used to identify the responses of the *PcNF-YA* genes to different nitrogen forms at different times (0, 6, 12 and 24 h) in *P.* × *canescens* roots (Figure 6). According to the gene expression profile shown in Figure 6, we found that *PcNF-YA12* was highly expressed under treatment with all three nitrogen forms. However, after nitrate treatment, the expression of *PcNF-YA12* increased gradually with increasing time. After the ammonium nitrate and ammonium treatments, the expression level of *PcNF-YA12* peaked at 6 h and then decreased over time. Additionally, compared with the results obtained at 0 h, the expression levels of *PcNF-YA1*, *PcNF-YA4*, *PcNF-YA6* and *PcNF-YA7* increased after nitrate treatment, while those of *PcNF-YA2*, *PcNF-YA3*, *PcNF-YA5*, *PcNF-YA8*, *PcNF-YA10* and *PcNF-YA11* decreased. The expression of *PcNF-YA9* first decreased and then increased. Compared with the results obtained at 0 h, the expression levels of all other *PcNF-YA* genes except for *PcNF-YA3* and *PcNF-YA6* increased after ammonium nitrate treatment. After ammonium treatment, the expression levels of all the *PcNF-YA* genes except for *PcNF-YA1* and *PcNF-YA6* increased compared with that at the 0 h time point.

### 2.9. MiR169d Expression Promoted Root Growth of Transgenic Poplar

To further explore the function of the miR169-mediated *PcNF-YA* genes in the response to different nitrogen forms, we constructed miR169d-overexpressing transgenic poplar plants. Nine independent transgenic lines were obtained. RT-qPCR verification showed that the expression of miR169d in transgenic miR169d plants significantly increased, while the expression of *PcNF-YA2* and *PcNF-YA6* decreased (Figure 7a, Appendix A). Additionally, three independent lines were selected for nitrate treatment, and the phenotype of each independent line was observed (Figure 7b). The adventitious root occurrence time of the transgenic poplar lines treated with nitrate for 10 days was earlier than that of the wild type, and the adventitious root number of the transgenic poplar lines treated with nitrate for 21 days was greater than that of the wild type (Figure 7b,c).

## 3. Discussion

NF-Y TFs are widely found in eukaryotes (yeast, plants and animals, etc.) [31,32]. NF-YA is the A subunit of the NF-Y complex, which is evolutionarily conserved but functionally diverse [9,19,21,28,33,34]. In this study, we identified a total of 13 *PcNF-YA* gene members in *P.* × *canescens* after strict screening and confirmation, which was different from the numbers in other species. These results suggest that the plant *NF-YA* gene family has developed redundancy and undergone functional differentiation during evolution to adapt to complex environments [4].

The phylogenetic relationships among NF-YA proteins in *P. × canescens* (Pc), *P. trichocarpa* (Ptr), *A. thaliana* (At), *O. sativa* (Os), *Pinus tabuliformis* (*Pita*) and *Prunus persica* L (Pp) were studied by bioinformatics analysis. The results showed that the PcNF-YA proteins of *P. × canescens* presented the highest similarity with the proteins of *P. trichocarpa*, followed by those of *Prunus persica*, and the lowest similarity with the proteins of angiosperms (*Pinus tabuliformis*). Moreover, we found that PcNF-YA2/6 and AtNF-YA3/8/5/6 clustered together, and previous studies have shown that *AtNF-YA3/5/6* respond to low nitrogen stress [29]. Therefore, we speculated that *PcNF-YA2/6* and *AtNF-YA3/5/6* in the same clade (Clade IV) may show similar biological functions in response to nitrogen nutrition. PcNF-YA7/8 and AtNF-YA2/10 clustered together, and studies have shown that *AtNF-YA2/10* can change the root architecture of *A. thaliana* and respond to nitrogen [29]. Therefore, we speculated that *PcNF-YA7/8* and *AtNF-YA2/10* were in the same clade (Clade I). They may present similar biological functions in changing the plant root architecture in response to nitrogen. This provides clues for future research on the biological mechanisms of *PcNF-YA* genes in the response to different nitrogen forms.

Conserved domains reflect the gene structure of a gene family [3]. The analysis of conserved domains in *P.* × *canescens* showed that all 13 members of the NF-YA family contained conserved NF-YB/C interaction regions and DNA binding regions. This suggested that they all belonged to the NF-YA family. Further multiple sequence alignment of the *P. × canescens* (Pc), *P. trichocarpa* (Ptr), *A. thaliana* (At), *O. sativa* (Os), *Pinus tabuliformis* (*Pita*) and *Prunus persica* L (Pp) proteins showed that the conserved domains of the NF-YAs of *P. × canescens* were similar to those of the proteins from these other species. The 3D structural homology modeling of *P. × canescens* PcNF-YAs also supported this conclusion, which indicated that the conserved domains of the 13 PcNF-YA members in *P. × canescens* were highly conserved among different species.

To better understand the biological functions of the PcNF-YAs, an exon-intron structure analysis of the *PcNF-YA* genes was performed and showed that the number of exons in most *PcNF-YA* genes was five. This finding was consistent with the results for *Brassica napus* [35], *S. tuberosum* [15], *R. communis* [14] and *Prunus persica* [11]. This evidence indicates that the structure of *PcNF-YA* genes is relatively stable and evolutionarily conserved. In addition to the similar gene structure, the conserved motifs of the PcNF-YA members that clustered together also showed great similarity. However, the gene structures and conserved motifs of different clades were different, and we speculate that different clades may have different functions. The results of the *PcNF-YA* gene structure and protein motif analyses further confirmed the reliability of the phylogeny.

The chromosome localization analysis of *PcNF-YA* family members revealed that the 13 members were randomly distributed nonhomogeneously on the six chromosomes of *P. × canescens*, which may be due to differences in chromosome structure and size. Fragment duplication and tandem duplication are the main drivers of plant gene family expansion [36] and promote the evolution and diversification of species [37]. The evaluation of the replication events of *PcNF-YA* genes in *P. × canescens* showed that tandem duplication and fragment duplication have occurred throughout the poplar genome, which is conducive to gene replication and gene family expansion. Furthermore, the assessment of the phylogenetic relationships and collinear gene pair events between different species showed that there might be another *NF-YA* gene with an unknown function in *P. × canescens* compared with *A. thaliana*, *O.sativa* and *Prunus persica*. Interestingly, we found that some *PcNF-YA* genes (*PcNF-YA2/6/7/8/5* and *11*) shared homologous events with all three selected species, implying that these genes existed before monocot and dicot divergence and may have played a key role in the evolution of the *PcNF-YA* family.

Previous studies have shown that *NF-YAs* are regulated by miR169 in many plants, including *A. thaliana* [5], *O. sativa* [13], *Populus* [38], *Brassica napus* L. [39], and wheat [21]. Based on previous small RNA sequencing results [30] and our BLAST analysis, we found that nine *PcNF-YA* genes contained miR169 binding sites, and miRNAs mainly targeted the 3’ UTR of *PcNF-YAs*, which was consistent with the results showing that conventional mRNAs usually bind to the 3’ UTR of target genes to block protein translation and regulate gene expression at the posttranscriptional level [40,41]. Furthermore, the identification of the interactions between miRNAs and *NF-YAs* will help us understand the roles of *PcNF-YA* gene family members under biotic or abiotic stress and at different developmental stages. Furthermore, it has been demonstrated that *NF-YA* genes involved in the response of herbaceous plants to nitrogen are regulated by miR169s [28,29]. Overexpression of miR169a in transgenic *A. thaliana* inhibited the expression of *AtNF-YAs* and reduced the amount of nitrogen that accumulated [29]. In *O. sativa*, overexpression of miR169 inhibited the expression of *NF-YAs* and promoted the growth of rice under low nitrogen treatment [28]. Our previous studies showed that nitrate promoted primary root elongation, inhibited lateral root initiation, induced miR169b expression and inhibited the expression of its target gene *PcNF-YA2* in poplar [30]. These studies indicate that the *NF-YA* genes are regulated by miRNAs and play important roles in plant growth and development and the response to abiotic stress.

We analyzed the tissue expression profile of the *PcNF-YA* gene family by RT-qPCR and found that the spatial and temporal expression patterns of the *PcNF-YA* gene family showed high heterogeneity. The expression levels differed among the roots, stem, leaves and bark, which may be related to their complex structure and motif composition. Additionally, the diverse tissue expression patterns revealed that *PcNF-YA*s are involved in many biological processes. However, most members of the *PcNF-YA* family in the same clade generally presented similar expression patterns in different tissues of *P. × canescens*.

*Cis*-acting elements that respond to phytohormones, as well as abiotic and biotic stress, and participate in plant growth and development were found at the promoter sites of *PcNF-YA* genes, indicating that *PcNF-YA*s play a potential role in regulating plant hormones, environmental stress responses and development. MYB and MYC elements accounted for the largest proportion of these elements. Previous studies have shown that MYB and MYC elements are involved in the ABA-dependent stress pathway under stresses such as drought, salinity and nutrient deprivation [42,43,44,45]. Based on the results of previous studies and the analysis of *cis*-regulatory elements in promoters [28,29,30], we used RNA sequencing and RT-qPCR to further verify whether *PcNF-YA* genes were involved in the response of poplar roots to different nitrogen forms. The results showed that several *PcNF-YA* genes responded to different nitrogen forms (Figure 5b). More interestingly, several *PcNF-YA* genes showed different or opposite expression patterns in response to different nitrogen forms, indicating that these *PcNF-YA* genes respond specifically to certain nitrogen forms and may play a unique role in improving the uptake and utilization of certain nitrogen forms in *P. × canescens* roots. Additionally, some homologous chromosome pairs showed similar expression patterns under treatment with the same nitrogen form, indicating that these genes (*PcNF-YA5/11* and *PcNF-YA7/8*) were not functionally differentiated during gene replication. However, there were some homologous genes that showed opposite expression trends. For example, the expression of *PcNF-YA4/12* was upregulated under treatment with all three nitrogen forms, while the expression trend of *PcNF-YA9* was significant under treatment with each of the three nitrogen forms, indicating that the functions of these genes have changed during evolution.

Subsequently, to further explore the function of the miR169-regulated *PcNF-YA* gene family in response to different nitrogen forms, miR169d was selected as the target for the generation of transgenic poplar material. After nitrate treatment, the rooting time of poplar overexpressing miR169d was earlier than that of wild-type poplar, and the number of adventitious roots was greater than that of wild-type poplar. We hypothesized that the overexpression of miR169 in transgenic poplar inhibited the expression of *PcNF-YAs* and promoted the growth and development of poplar roots under nitrate treatment. This result is consistent with the results obtained in rice. The overexpression of miR169 in transgenic rice inhibited the expression of *NF-YAs* and promoted the growth of rice under low nitrogen treatment [28]. Our previous studies have also shown that nitrate can promote primary root elongation and inhibit lateral root initiation in poplar, while nitrate treatment induces miR169b expression and inhibits *PcNF-YA*2 expression [30]. These results suggest that *NF-YA* gene family members can respond to nitrogen in plants and that this biological process may be regulated by miR169.

## 4. Materials and Methods

### 4.1. Sequence Retrieval of NF-YA Family Members in P. × canescens

To identify the complete NF-YA family of *P. × canescens*, a genome-wide BLASTP search of *P. × canescens* (https://www.aspendb.org/index.php/databases/spta-717-genome (accessed on 25 December 2020) was performed using the *AtNF-YA* genes from *Arabidopsis thaliana* (https://www.Arabidopsis.org/ (accessed on 25 December 2020) and the *PtNF-YA* genes from *P. trichocarpa* (https://phytozome.jgi.doe.gov/pz/portal.htm (accessed on 25 December 2020)). Subsequently, the Pfam hidden Markov model (PF02045) was used to determine whether the candidate genes contained a specific conserved CBF_NF-YA domain using HMMER package version 3.0. ExPASy (https://web.expasy.org/protparam/ (accessed on 11 May 2022)) was used to obtain the gene sequence ID, chromosomal location, number of amino acids, molecular weight, theoretical pI and other information for each NF-YA in *P. × canescens*. WoLF PSORT II (https://www.genscript.com/wolf-psort.html?src=leftbar (accessed on 11 May 2022)) was used to predict the subcellular localization of genes.

### 4.2. Phylogenetic, Multiple Sequence Alignment, Gene Structure and Protein Motif Analyses of PcNF-YA Proteins

Important information about *NF-YA* gene families in the *P. trichocarpa* (Ptr), *A. thaliana* (At), *O. sativa* (Os), *Pinus tabuliformis* (*Pita*) and *Prunus persica* (Pp) genomes was downloaded from the Phytozome and NCBI databases. These include protein sequences and CDSs. ClustalX (http://www.clustal.org/clustal2/ (accessed on 8 June 2022)) was used to compare the *PcNF-YA* gene family of *P. × canescens* with those of other species.

The phylogenetic tree was constructed via the NJ method with 1000 bootstrap replications using MEGA 7.0.21 software [46]. Based on the CDSs and full-length gene sequences of the *PcNF-YA* genes in the *P. × canescens* database, the exon-intron structure of the *PcNF-YA* genes was analyzed with GSDS software (http://gsds.cbi.pku.edu.cn/, 2.0 (accessed on 28 May 2022)). Based on the PcNF-YA protein sequence in the *P. × canescens* database, the conserved protein motifs were predicted with the MEME website (http://meme.nbcr.net/meme (accessed on 11 May 2022)). Each of the PcNF-YA protein sequences showed an E-value less than 10, and the motif matches shown presented position *p* values of less than 0.0001 [47]. Additionally, sequence alignment was performed with DNAMAN software to identify the motifs of the PcNF-YA gene interaction region and DNA binding region. Then, the tertiary structure of the PcNF-YA family proteins was predicted at the Phyre^2^ website (http://www.sbg.bio.ic.ac.uk/phyre2/html/page.cgi?id=index (accessed on 16 June 2022)).

### 4.3. Chromosomal Distribution and Synteny Analysis of PcNF-YA Proteins

The chromosomal locations of the *PcNF-YA* gene family members were visualized by using MapChart software [48]. Multiple Collinear Scan Toolkit X (MCScanX) was used to scan *PcNF-YA* gene collinearity and analyze gene replication events [49]. TBtools was used to perform an analysis of the synteny of *P. × canescens* with *A. thaliana*, *O. sativa* and *Prunus persica* [50].

### 4.4. Identification of Cis-Regulatory Elements in Promoter Regions

The 2 kb sequences upstream of *PcNF-YA* genes were obtained as the promoter region based on the *P. × canescens* database, and the online software PlantCARE (http://bioinformatics.psb.ugent.be/webtools/plantcare/html/ (accessed on 6 June 2022)) was used to identify *cis*-elements in the promoter sequence.

### 4.5. Prediction of PcNF-YAs Targeted by miR169

The potential miR169 family members in *P. × canescens* roots were explored based on previous sequencing results [30]. Then, the sequences of the poplar miR169 family members were downloaded from the miRBase database (https://www.mirbase.org/ (accessed on 26 June 2021)), and the previously reported Pc-miR169 sequence was compared with the poplar miR169 sequences from the miRBase database to remove sequences with a mismatch rate greater than 3. The sequence was identified as a member of the *P. × canescens* miR169 family. Finally, using TAPIR tools (http://bioinformatics.psb.ugent.be/webtools/tapir/ (accessed on 16 June 2022)), the cDNAs of the *P.* × *canescens PcNF-YA* gene and mature miR169 were analyzed, and the possible binding sites of Pc-miR169 and *PcNF-YA* were predicted.

### 4.6. Plant Materials, Growth Conditions, and Different Nitrogen Form Treatments

Hydroponically grown seedlings of *P.* × *canescens* were used as the experimental materials. To examine the expression of the 13 *PcNF-YA* genes in different tissues of *P.* × *canescens*, we obtained root, stem, leaf and bark samples of *P.* × *canescens* by tissue culture for 1 month followed by hydroculture for 2 months. The culture methods were performed as described by Zhou and Wu [30]. To obtain sufficient amounts of samples, we mixed 3 samples of the same amount of each tissue in a pool as one test sample and repeated the experiment 3 times. The obtained samples were immediately frozen in liquid nitrogen at −80 °C.

To examine the expression of the 13 *PcNF-YA* genes in *P.* × *canescens* roots under different nitrogen treatments, nitrogen treatments [1 mM nitrate (S1), 500 μM ammonium nitrate (S2) and 1 mM ammonium (S3)] were performed according to Zhou’s method [30]. Raw RNA-seq data from plants treated with different nitrogen forms for 21 days were obtained in our previous study [30] and deposited in the NCBI database (accession number: PRJNA631840). Low-quality sequences were excluded, and *PcNF-YA* family members were identified from the RNA-seq data for transcriptome enrichment analysis. HISAT2 software (Baltimore, MD, USA) [51] was used to compare reads from all samples with the *P.* × *canescens* genome database. Transcriptome abundance was quantified under treatment with different nitrogen forms, and *PcNF-YA* gene expression levels were identified according to the fragments per kilobase per million mapped fragments (FPKM) method. Cytoscape software (3.5.1, Seattle, WA, USA) was used to draw heatmaps of the gene expression levels of each *PcNF-YA* family member. Additionally, root samples were collected after 0, 6, 12 and 24 h of treatment with different nitrogen forms. Three root tissue samples of equal amounts were pooled as one test sample, and the experiment was repeated 3 times. The obtained test samples were frozen with liquid nitrogen and stored at −80 °C for RT-qPCR analysis.

### 4.7. Total RNA Extraction and RT-qPCR

The RNA extraction and reverse transcription methods were performed as described by Zhou et al. [52]. Total RNA was extracted with a total RNA extraction kit (TRK1001, LianChuan (LC) Science, Hangzhou, China), and a reverse transcription kit (PrimeScript^TM^ RT reagent Kit, Dalian, China) was used for reverse transcription to cDNA. RT-qPCR analysis was performed for each gene in the *PcNF-YA* family under different nitrogen treatment conditions at different times, and the steps and reagents used were as described by Zhou et al. [53]. The specific primers for the *PcNF-YA* family were designed using Primer 3 (https://bioinfo.ut.ee/primer3-0.4.0/ (accessed on 17 January 2021)) and are listed in Appendix A. Each sample was analyzed with three biological replicates. The expression characteristics of the *PcNF-YA* genes under treatment with different nitrogen forms and at different times were analyzed. The relative expression levels were calculated using the 2*^−ΔΔCt^* method. The data for the Ct values obtained by RT-qPCR were also tested for normality prior to statistical analysis using one-way analysis of variance (ANOVA), with different time points as a factor. Differences between means were considered significant when *p* < 0.05 according to the ANOVA F test.

### 4.8. Populus Transformation

The precursor sequence of *P.* × *canescens* miR169d was transformed into poplar using the method described by Wen [54]. Transgenic poplar plants carrying the miR169d precursor sequence were tested at the DNA level, and the expression level was verified at the RNA level for further study. Through the tissue culture and propagation method [54], transgenic poplar plants and wild-type plants were subcultured in LA medium containing 1 mM nitrate at the same time, and the root phenotype was observed after 10 and 21 days.

## 5. Conclusions

In summary, *PcNF-YA* genes play an important role in plant development and stress responses, but their roles in the response to nitrogen in plants need to be further studied. We identified 13 *PcNF-YA* genes and analyzed the characteristics of these gene family members based on their conserved amino acid sequence domains, phylogenetic development, intron and exon structure, homology, collinearity, predicted miRNA targets and expression patterns. In addition, a comprehensive bioinformatics analysis and experimental verification demonstrated that the *PcNF-YA* genes showed specific expression patterns under treatment with different nitrogen forms and revealed that the *NF-YA* gene family was involved in the responses of poplar roots to different nitrogen forms, thus indicating their potential roles in regulating root growth and development. These results provide a basis for understanding the molecular mechanism of the plant response to nitrogen.

## Figures and Tables

**Figure 1 ijms-23-11217-f001:**
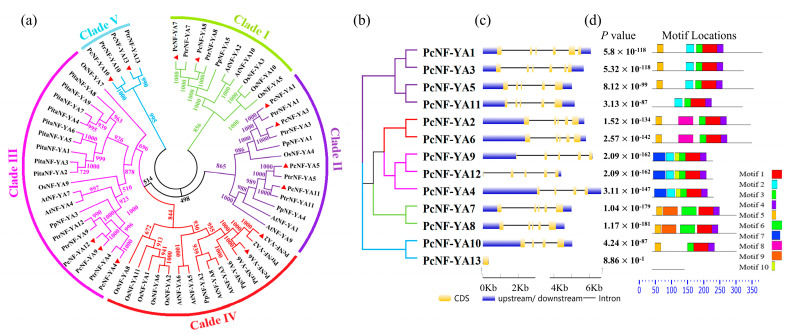
Phylogenetic, gene structure and protein motif analyses of NF-YA proteins in *P.*
*× canescens. P.*
*× canescens* (Pc), *P. trichocarpa* (Ptr), *A. thaliana* (At), *O. sativa* (Os), *Pinus tabuliformis* (*Pita*) and *Prunus persica* L (Pp). Green, purple, pink, red and blue branches represent Clades I, II, III, IV and V, respectively. (**a**) PcNF-YA proteins are marked with red filled triangles. (**b**) The amino acid sequences of PcNF-YAs were used to create a phylogenetic tree with the neighbor-joining (NJ) method. Bootstrap = 1000. Clades I to V are displayed with different colors. (**c**) The intron/exon structures of the PcNF-YA family. Blue rectangles, yellow rectangles and black lines indicate UTRs, exons and introns, respectively. (**d**) Conserved motif analysis of PcNF-YA proteins. Conserved motifs shown in different colored boxes were created by MEME, and sequence information for each motif is provided in Appendix A. The scales at the bottom in (**c**,**d**) represent the sequence lengths.

**Figure 2 ijms-23-11217-f002:**
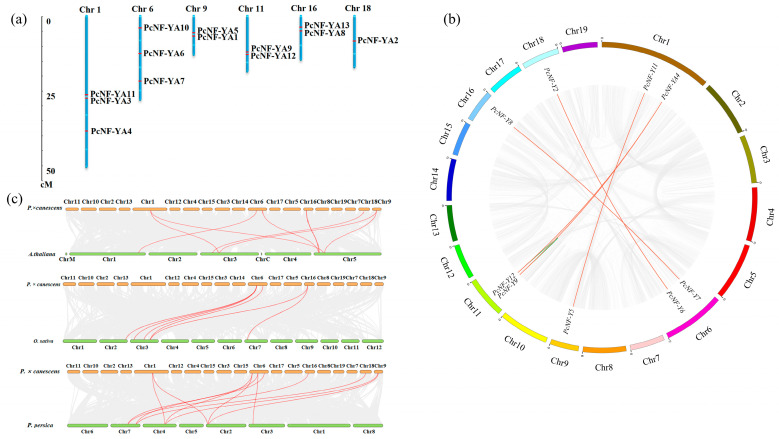
Chromosome localization, gene duplication and synteny analysis of *PcNF-YA* genes. (**a**) Characteristics of 13 genes with PcNF-YA domains identified in the genome of *P.* × *canescens*. The scale on the left represents the lengths (Mb, megabases) of the *P.* × *canescens* chromosomes. (**b**) *PcNF-YA* gene segmental duplication analysis. The gray lines represent the synteny blocks in *Populus* among chromosomes, duplicated *PcNF-YA* gene pairs are shown by five thick red lines, and tandem duplication gene pairs are shown by one green line. The nineteen chromosomes are displayed on the map in a circular pattern. (**c**) Synteny analysis related to *NF-YA* genes between the genome of *P.* × *canescens* and those of *A. thaliana*, *O. sativa* and *Prunus persica*. The light gray lines represent the collinear blocks between *Populus* and *Arabidopsis*, *Populus* and rice, and *Populus* and peach. The syntenic *PcNF-YA* gene pairs are shown connected by the red lines.

**Figure 3 ijms-23-11217-f003:**
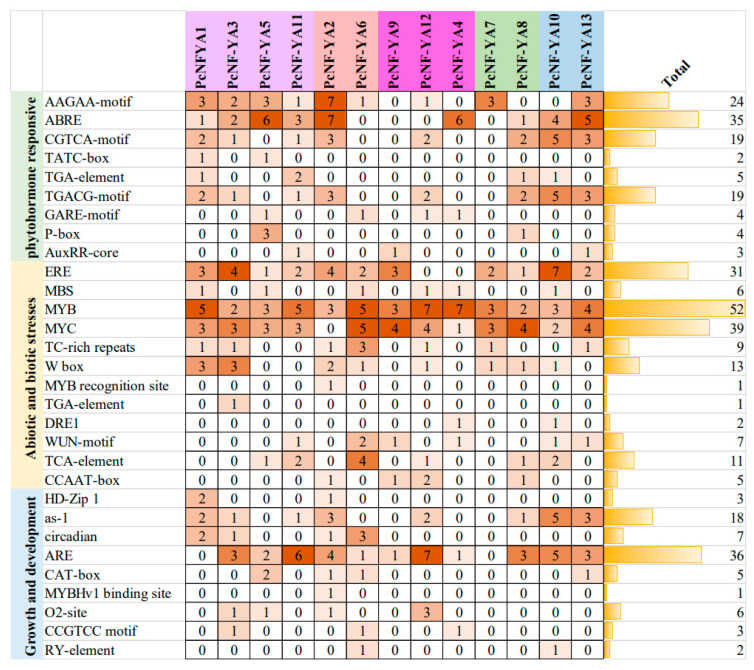
*Cis*-elements of the *PcNF-YA* gene family in the genome of *P.* × *canescens*. *PcNF-YA* family members are listed on the X axis, purple, red, pink, green and blue represent different clades of the *PcNF-YA* family. All *cis*-elements in promoter regions are listed on the Y axis. The figure indicates the number of *cis*-elements. The orange bars represent the total number of occurrences of these *cis*-elements in the 13 *PcNF-YA* family members.

**Figure 4 ijms-23-11217-f004:**
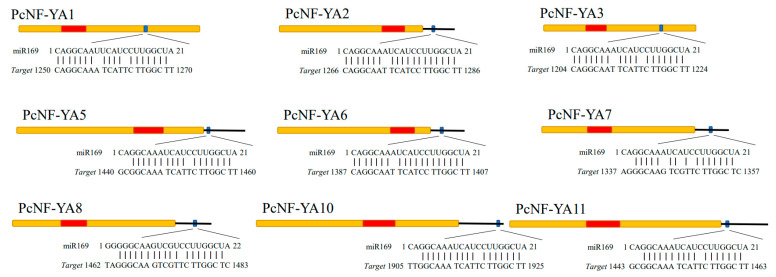
miR169 family members and their target site in the *PcNF-YA* genes. miR169 target site in the *PcNF-YA1*, *2*, *3*, *5*, *6*, *7*, *8*, *10* and *11* genes. Yellow boxes represent the CDS, red boxes represent the CBF domain, and lines represent the 3′ UTR. The miR169 target sites with the nucleotide positions of *PcNF-YA* transcripts are shown in blue.

**Figure 5 ijms-23-11217-f005:**
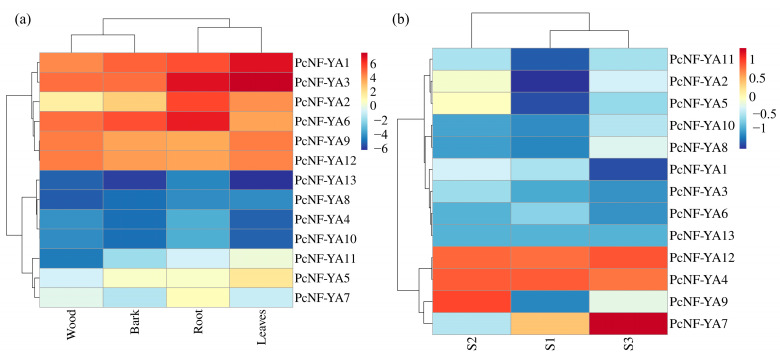
Expression pattern analysis of *PcNF-YA* genes in different tissues and under different nitrogen form treatments. (**a**) The expression level of each *PcNF-YA* in different tissues was compared with the median expression level of all genes, and the log_2_ (fold change) in *PcNF-YA* expression was calculated for heatmap analysis. (**b**) Transcriptome analysis of the *PcNF-YA* gene expression profile under different nitrogen form treatments at 21 days. The log_2_ (fold change) in *PcNF-YA* expression was calculated for heatmap analysis. S1, S2 and S3 represent the nitrate treatment, ammonium nitrate treatment and ammonium treatment, respectively.

**Figure 6 ijms-23-11217-f006:**
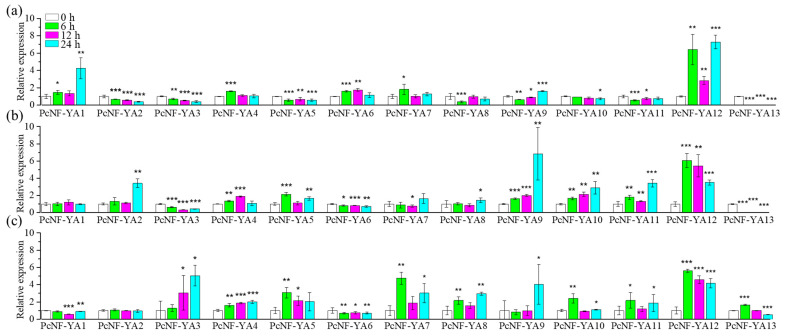
Expression pattern analysis of *PcNF-YA* genes under different nitrogen form treatments at different times by RT– qPCR. (**a**) Expression profile analysis of the *PcNF-YA* genes at different times under nitrate treatment; (**b**) Expression profile analysis of the *PcNF-YA* genes at different times under ammonium nitrate treatment; (**c**) Expression profile analysis of the *PcNF-YA* genes at different times under ammonium treatment. The data are expressed as the mean ± SD (*n* = 3). *P*-values from one-way ANOVA for the different times are indicated: *: *p* < 0.05; **: *p* < 0.01; ***: *p* < 0.001.

**Figure 7 ijms-23-11217-f007:**
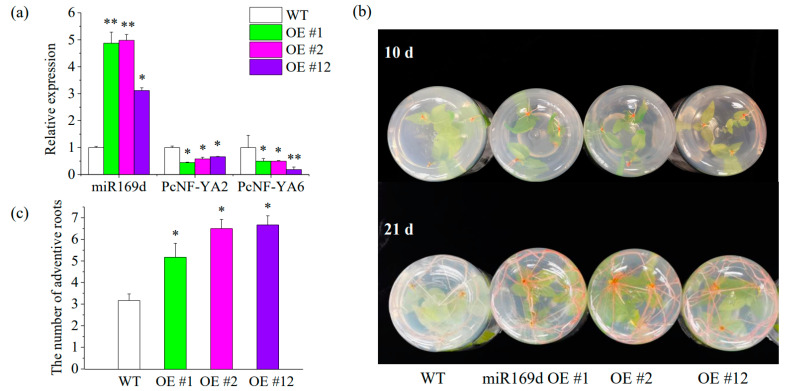
Phenotypic analysis of transgenic *Populus* with the miR169d gene under nitrate treatment. (**a**) Expression analysis of the miR169d and *PcNF-YA2*/*6* genes with RT-qPCR in transgenic *Populus* lines; (**b**) Root phenotypes of wild-type (WT) and transgenic *Populus* lines (miR169d OE #1, OE #2 and OE #12) after 10 and 21 days of nitrate treatment; (**c**) the adventitious root numbers of WT and transgenic *Populus* lines (miR169d OE #1, OE #2 and OE #12) after 10 and 21 days of nitrate treatment. *p*-values from one-way ANOVA for the different lines are indicated: *: *p* < 0.05; **: *p* < 0.01.

**Table 1 ijms-23-11217-t001:** *NF-YA* genes identified in *P.* × *canescens*.

Gene Name	Gene ID	Chr	The Number of Introns	The Number of Exons	Amino Acid Length (aa)	Protein Mol. Wt. (kDa)	pI	Subcellular Localization
*PcNF-YA1*	potri.009G060600	9	5	6	377	41.31805	7.83	Nucleus
*PcNF-YA2*	potri.018G064700	18	4	5	337	37.23665	8.94	Nucleus
*PcNF-YA3*	potri.001G266000	1	5	6	313	34.32908	8.84	Nucleus
*PcNF-YA4*	potri.001G372100	1	3	4	210	23.21267	7.13	Nucleus
*PcNF-YA5*	potri.009G052900	9	4	5	347	38.17855	6.35	Nucleus
*PcNF-YA6*	potri.006G145100	6	4	5	341	37.15937	9.05	Nucleus
*PcNF-YA7*	potri.006G201900	6	4	5	323	35.00364	9.26	Nucleus
*PcNF-YA8*	potri.016G068200	16	4	5	319	34.99273	9.28	Nucleus
*PcNF-YA9*	potri.011G101000	11	4	5	208	22.83747	9.15	Nucleus
*PcNF-YA10*	potri.006G053500	6	3	4	300	33.37344	9.42	Nucleus
*PcNF-YA11*	potri.001G257600	1	4	5	309	33.92788	8.76	Nucleus
*PcNF-YA12*	potri.011G098400	11	4	5	208	22.83747	9.15	Nucleus
*PcNF-YA13*	potri.016G054100	16	0	1	53	5.802	11.47	Nucleus

## Data Availability

All of the small RNA sequences are available in the Sequence Read Archive (SRA) under project ID PRJNA631845 (https://www.ncbi.nlm.nih.gov/bioproject/PRJNA631845). The transcriptomic sequencing data were also submitted to the SRA under project ID PRJNA631840 (https://www.ncbi.nlm.nih.gov/bioproject/PRJNA631840).

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
