# Peer review of "Genome-Wide Identification and Characterization of the NF-YA Gene Family and Its Expression in Response to Different Nitrogen Forms in Populus × canescens"

_ijms, 2022, doi:10.3390/ijms231911217_

Round 1
Author Response
General answer: Thank you for taking your time to review our paper. Your recognition, encouragement and comments on our manuscript are highly appreciated. Please refer to our answers to your comments below. I hope that you are satisfied with our answers.
- It would be significant to obtain the information of the whole NF-Y TFs, including NF-YAs, -YBs and -YCs in Populus × canescens. The authors could then concentrate on the NF-YA subunit, as described in the article. Thus, in silico analyses of the NFY TFs in Populus × canescens are highly recommended.
Answer: Thank you for your constructive comment.
It is important to obtain the information of the whole NF-Y TFs, but in this paper, only NF-YAs of Populus × canescens were studied for the following reasons: first, as described in the paper, despite the abundance of NF-YB and NF-YC subunits, NF-YA restricts heterotrimer formation and subsequent DNA binding. Therefore, the roles of NF-YA are particularly critical. Second, the previous analysis of sequencing data of different nitrogen forms showed that PcNF-YAs responded to different nitrogen forms in poplar roots. Meanwhile, we speculated that PcNF-YAs in poplar roots respond to different nitrogen forms most likely through the regulation of miR169s. However, were not identified as responding to different nitrogen forms in P. canescens and were not regulated by miRNAs. This is consistent with reports in Arabidopsis (Zhao, et al. New Phytologist, 2011) and rice (Yu, et al. Plant and cell physiology, 2018). In summary, this paper focuses on the identification of NF-YA gene family members and the regulation of PcNF-YAs in response to different nitrogen forms through miR169s in P. × canescens. This also lays a foundation for our subsequent study on the function of miR169-NFYA modules. Therefore, this paper does not carry out silico analysis on NF-YB and NF-YC subunits.
- Authors state that 2 Kb sequences upstream to transcription start site were retrieved for the prediction of cis regulatory elements. How did transcription start site get determined? Furthermore, the PlantCARE is out of date because the authors had to manually search for many well characterized stress-responsive cis-acting elements, particularly nitrate-responsive cis elements. As a result, the Results and Discussion section related to this part should concentrate on the stress- and hormone- responsive cis-regulatory elements.
Answer: Thank you for your constructive comment.
The transcription start site was determined by predicting the CDS using TBtools software, and the sequence information of NF-YA family members in P. trichocarpa sequence with well-annotated information was also referenced. The PlantCARE database is widely accepted by researchers as a cis-acting element prediction website. For example, Zhang et al., International Journal of Molecular Sciences, 2022; Zang et al., International Journal of Molecular Sciences, 2022; Zhang et al., Frontiers in Plant Science, 2022 and Muhammad Mubashar Zafar et al., BMC Plant Biology, 2022 previously used this online analysis software to predict cis-acting element of genes. Meanwhile, in the Results and Discussion section, we indicated that the MYB and MYC elements were analyzed as stress response elements and that they may participate in the process of plant response to nitrogen by participating in the ABA-dependent stress pathway.
– The PlantTFDB database is a well-known server that contains detailed information on all TFs found in plant species. Have the authors checked the NF-Y TFs in this database, possibly the NF-YA subunit in Populus canescens? Is your genome-wide search any different?
Answer: Thank you for your constructive comment.
We have searched for NF-YA TFs in the PlantTFDB database and found that our results are similar to those for P. trichocarpa (13 PtrNF-YAs), with more NF-YA TFs compared to P. euphratica (9 PeNF-YAs). However, PcNF-YA TFs were not included in the database, which indicates that the amount of NF-YA TFs is different across poplar varieties. For this reason, we examined NF-YA TFs in P.× canescens in the present study.
- A method for predicting PcNF-YA subunit subcellular localization is still lacking. Please describe this method in the appropriate place, if possible.
Answer: Thank you for your careful reading.
We have added the method for predicting PcNF-YA subunit subcellular localization. The sentence " WoLF PSORT II (https://www.genscript.com/wolf-psort.html?src=leftbar) was used to predict the subcellular localization of genes. " has been added to the Materials and Methods. Lines 418 to 420.
- The relative expression analysis in RT-qPCR validation is extremely limited. If possible, the authors should include more details in their methods, such as student's ttest, *p < 0.05, or how they use T-test on RT-qPCR values.
Answer: Thank you for your constructive comment.
We have added more details in the Materials and Methods. The sentence " The data for the Ct values obtained from RT-qPCR were also tested for normality prior to statistical analysis using One-way analysis of variance (ANOVA), with different time points as a factor. Differences between means were considered significant when P < 0.05 according to the ANOVA F-test." has been added to the Materials and Methods. Lines 418 to 420.
- Please check for grammatical and spelling mistakes as some in the pdf file. Many words should be italicized. Some duplicated words should be noted to check.
Answer: Thank you for your careful reading.
We have corrected these grammatical and spelling mistakes.

Reviewer 2 Report
Authors identified and characterized the NF-YA gene family members of P × canescens at the whole-genome level. Also, miRNA targets and promoter cis-acting elements of the NF-YA genes were identified, and tissues specific expression was analysed. Functional analysis of PcNF-YA genes revealed their role in root growth and development.
· L. No. 94-95; Add the related figures to support the statement; Subcellular localization analysis showed that PcNF-YA was located in the nucleus.
· L. No. 263; Add the statistical information in figure 6
· It is more useful if you present more information about the Populus transformation. In particular, plant generation and segregation details
Author Response
General answer: Thank you for your recommendation and suggestions. We have made the relevant revisions, and I hope that you can recognize our efforts on this revised version. We have provided point- by- point answers to the comments in the following section.
- L. No. 94-95; Add the related figures to support the statement; Subcellular localization analysis showed that PcNF-YA was located in the nucleus.
Answer: Thank you for your careful reading.
We have added the method for predicting PcNF-YA subunit subcellular localization. The sentence " WoLF PSORT II (https://www.genscript.com/wolf-psort.html?src=leftbar) was used to predict the subcellular localization of genes. " has been added to the Materials and Methods. Lines 418 to 420.
- L. No. 263; Add thestatistical information in figure 6
Answer: Thank you for your constructive comment.
We have added the statistical information in figure 6.
- It is more useful if you present more information about the Populus transformation. In particular, plant generation and segregation details
Answer: Thank you for your constructive comment.
The validation of NF-YA genes function by Populus transgenic technology is the focus of our next study, but this aspect is not the focus of this manuscript. This study focused on the genome-wide identification of the NF-YA gene family in P.× canescens and its response to different nitrogen forms. This study lays the foundation for subsequent functional verification. Therefore, we do not have more information about the Populus transformation in this article.
